# Determination of Nitrate and Nitrite in Swiss Chard (*Beta vulgaris* L. subsp. *vulgaris*) and Wild Rocket (*Diplotaxis tenuifolia* (L.) DC.) and Food Safety Evaluations

**DOI:** 10.3390/foods11172571

**Published:** 2022-08-25

**Authors:** Marco Iammarino, Giovanna Berardi, Valeria Vita, Antonio Elia, Giulia Conversa, Aurelia Di Taranto

**Affiliations:** 1Department of Chemistry, Istituto Zooprofilattico Sperimentale della Puglia e della Basilicata, 71121 Foggia, Italy; 2Department of Agriculture Sciences, Food, Natural Resources and Engineering, University of Foggia, 71122 Foggia, Italy

**Keywords:** margin of safety, nitrate, nitrite, risk exposure, Swiss chard, wild rocket salad

## Abstract

Nitrite and nitrate levels in vegetables are a matter of concern due to their toxicity at high levels and nitrate high accumulation. Moreover, there is a lack of knowledge about their levels in some types of widely consumed vegetables such as chard and rocket. In this study, 124 Swiss chard and wild rocket samples were analyzed for determining nitrite and nitrate using validated and accredited analytical methods by ion chromatography with conductivity detection. High nitrite concentrations, up to 219.5 mg kg^−1^ f.w., were detected in one Swiss chard and three wild rocket samples. One Margin of Safety (MoS) value was <1. Regarding nitrate, in Swiss chard samples the mean concentration (2522.6 mg kg^−1^) was slightly higher than those reported in the literature for spinach and lettuce. No MoS was <1, but 83% of values were <100. Nitrate concentrations higher than the legal limit were quantified in 11 rucola samples. The verification of 25% of wild rocket samples with nitrate concentration higher than the legal limit confirmed the need for official control. This study also suggests the introduction of legal limits for nitrite/nitrate in Swiss chard and nitrite in wild rocket.

## 1. Introduction

Nitrate and nitrite are two compounds present in the environment, since they take part in the nitrogen cycle. Atmospheric nitrogen may also be oxidized to nitrate and nitrite by microorganisms in soil, plants, and water [1,2].

In agriculture, nitrate and its salts are widely used mainly as inorganic fertilizers, but these compounds are also used in other fields, i.e., as food preservatives, oxidizing agents, in the chemical industry, explosives preparation, etc. Due to this high anthropic impact and fossil fuel combustion, during the second half of the 20th century, nitrate accumulated in the environment drastically [1,2,3,4].

Two meaningful documents focusing on nitrite/nitrate occurrence in foods and their toxicity were published by the European Committee of experts in the 1990s [5,6]. More recently, the European Food Safety Authority (EFSA) released two scientific opinions regarding nitrite/nitrate levels in food and feed and their implications for human health [7,8]. The undesired presence of nitrite in animal feed was also discussed by the EFSA in a scientific opinion published in 2009 [9]. This last document identified, other than the well-known methemoglobinemia concern [10], two other toxicological endpoints related to nitrite toxicity, namely the equivocal evidence for carcinogenesis in female mice and the adrenal zona glomerulosa hypertrophy in rats.

Another important aspect of food safety related to nitrite intake is the possible formation of N-nitrosamines, a class of pro-carcinogenic compounds considered to be causally involved in the development of cancer in humans and animals, as first proposed by Barnes and Magee [11]. These compounds may also be found in food as the consequence of reaction between nitrosating agents with amino-based substances [12,13]. The most common nitrosating agents of food are nitrite salts and nitrogen oxides. Although the most representative type of food linked to this risk is processed meat [14], several authors also reported the presence of different N-nitrosamines in vegetables, such as pickled and fermented vegetables. More in-depth, these authors quantified levels of N-nitrosodimethylamine and N-nitrosopyrrolidine at concentrations up to 15 and 25.5 µg kg^−1^, respectively [15,16,17,18]. These products contained no added nitrite, but were characterized by high levels of natural nitrate, possibly explaining the formation of N-nitrosamines by the reduction of nitrate to nitrite under the acidic conditions present in these products.

It is also important to underline that several authors demonstrated some health benefits of dietary nitrate supplementation in humans, especially elderly, so that the discussion about the risk/benefit of nitrate in the scientific community is still fervent [19,20].

Vegetables, particularly leafy vegetables, are characterized by high nitrate concentration, so that a great part of nitrate daily intake derives from vegetable consumption [9,20]. This nitrate accumulation, which takes place especially in the leaves, is mainly due to the increasing use of nitrogen fertilizers in agriculture [21], with the residual levels depending on many factors, such as the plant genotype, temperature, light exposure, cultivation system [22,23] water relations, carbon dioxide concentration, soil type, and contemporary use of herbicides [24,25]. Regarding nitrites, this compound can be present in vegetables only at a low concentration, derived from endogenous nitrate reduction [5]. However, Iammarino et al. reported interesting findings regarding the possible presence of high levels of nitrite in leafy vegetables (up to 197.5 and 66.5 mg kg^−1^ f.w. in spinach and lettuce, respectively) [26].

The nitrate accumulation in these types of vegetable is also worthy of investigation in food safety programs. Different studies have reported nitrate concentrations in spinach and lettuce samples higher than the respective legal limits [27,28], while few data are available related to other widely consumed leafy vegetables such as Swiss chard (*Beta vulgaris* L. subsp. *vulgaris*) and wild rocket (*Eruca sativa* and *Diplotaxis* spp.) [29,30,31].

From the legislative point of view, nitrite was included in the European Directive No. 2002/32/EC on undesirable substances in animal feed. The maximum nitrite limit (expressed as NaNO_2_) in fish meal and complete feeding stuff (excluding feeding stuff intended for pets except for birds and aquarium fish) is equal to 60 and 15 mg kg^−1^, respectively [32]. Regarding vegetables, no nitrite maximum level has been established so far. This means that nitrite should be absent (<method LOQ) in leafy vegetables [33]. The European Commission Regulation (EU) No. 1258/2011 only established the legal limits for nitrate relating to some types of leafy vegetables, such as spinach, lettuce, and rucola (*Eruca sativa*, *Diplotaxis* spp., *Brassica tenuifolia*, *Sisymbrium tenuifolium*), processed cereal-based foods, and baby foods for infants and young children. Thus, the legal limits established for nitrate in wild rocket correspond to 7000 and 6000 mg kg^−1^ f.w., if harvested from 1 October to 31 March and from 1 April to 30 September, respectively. Strangely, no maximum permitted level has been established for Swiss chard so far.

In this study, the nitrite and nitrate levels that characterize Swiss chard (*Beta vulgaris* L. subsp. *vulgaris*) and wild rocket (*Diplotaxis tenuifolia* (L.) DC.) samples, as collected on the market, were investigated. The nitrate concentrations were compared to those related to the most studied leafy vegetables, such as spinach and lettuce, and to the legal limits available for wild rocket. The possible quantification of nitrite, also at high levels, was monitored as well, to confirm its presence, analogously to that already reported for spinach and lettuce [34]. The analyses were carried out by using two fully validated and accredited methods of ion chromatography with conductivity detection.

## 2. Materials and Methods

### 2.1. Sample Collection and Preparation

In total, 124 samples composed of fresh Swiss chard (*Beta vulgaris* L. subsp. *vulgaris*, *cicla* group) (23 samples), fresh large ribbed Swiss chard (*Beta vulgaris* L. subsp. *vulgaris*, *flavescens* group) (13 samples), frozen and fresh-cut Swiss chard, and fresh and fresh-cut wild rocket salad (*Diplotaxis tenuifolia* (L.) DC.) (22 samples each) (Figure 1) were randomly collected during 2021–2022 from local stores in Foggia (Italy). Most samples originated from the same territory of collection (50% of chard and 50% rocket), though the origin of the remaining samples was from different Italian regions: Apulia (17.5% of chard and 18% of rocket), Campania (10% of chard and 27.5% of rocket), Emilia-Romagna (15% of chard), Veneto (2.5% of chard), Abruzzo (5% of chard), and Marche (4.5% of rocket). All samples were cultivated by conventional procedures. On arrival at the laboratory of the Chemistry Department of Istituto Zooprofilattico Sperimentale della Puglia e della Basilicata (Foggia, Italy), the fresh and fresh-cut samples were stored at 4 °C (±2 °C) and analyzed within 24 h to avoid any physico-chemical changes in the products. Frozen samples were stored at −18 °C (±2 °C) and analyzed within 1 week.

The whole sample was completely homogenized by using a specific mixer (BUCHI B-400, BUCHI Italia s.r.l., Assago, Milan, Italy). The size of each sample varied from 300 to 600 g for chard and from 100 to 200 g for rocket. Two grams (±0.01 g) of the homogenized sample were weighed in a 500 mL glass flask, together with 200 mL of ultrapure water. The sample was placed in a hot bath (70 °C ± 4 °C) for 5 min, shaking in the meanwhile. After cooling, ~5 mL of the solution were filtered using 0.22 µm syringe filters for ion chromatography (Sartorius AG, Goettingen, Germany), and a final dilution with ultrapure water was made to re-enter in the calibration range, if needed. Each sample was analyzed twice, and the result was expressed as the mean of two measurements.

### 2.2. Chemicals and Working Standard Solutions

Anhydrous sodium carbonate (>99.5%) was purchased from VWR International s.r.l. (Milan, Italy, while the nitrite and nitrate certified anion standard solutions for ion chromatography (1000 mg L^−1^) were supplied by Sigma-Aldrich (Stenheim, Germany). The solvent used for standards and mobile phase preparation and sample extraction was ultrapure water, with a specific resistance of 18.2 MΩ cm^−1^, supplied by an Arium mini essential UV system (Sartorius AG, Goettingen, Germany). The sodium carbonate solution, used as the mobile phase, was filtered before use through 0.45 μm membrane filters and degassed with helium. The stock solutions of nitrite and nitrate at a concentration of 1000 mg L^−1^ were opportunely diluted for calibration purposes of the traditional ion chromatography method (HPIC) to provide nitrite concentrations of 0.2, 0.4, 0.8, 1.6, and 6.5 mg L^−1^ and nitrate concentrations of 0.6, 1.2, 2.4, 4.8, and 12.5 mg L^−1^. The calibration levels used for the confirmatory technique, capillary ion chromatography (CIC), were 0.02, 0.1, 0.2, 2.0, and 20 mg L^−1^ for both nitrite and nitrate.

### 2.3. Apparatus and Chromatographic Methods

The chromatographic determinations of nitrite and nitrate were accomplished by using a high-pressure ion chromatography system (Thermo Scientific Dionex ICS-6000 HPIC System, Thermo Fisher Scientific Inc., Waltham, MA, USA) composed of a gradient mixer (Dionex GM-4, 2 mm), an SP Single Pump (ICS-6000), a Dionex anion self-regenerating suppressor (ADRS 600, 4 mm) set at the recommended voltage, a DC detector set to conductivity mode, and an injection valve with a 25-µL loop. The column compartment temperature was set at 20 °C, and the chromatographic column used was the IonPac AS9-HC (250 mm × 4 mm i.d., particle size: 9 µm), coupled to AG9-HC pre-column (Thermo Fisher Scientific Inc.). The isocratic chromatographic separation is well-known and standardized for meat-products’ analysis, and subsequent ruggedness studies confirmed its applicability for vegetables’ analysis [34,35,36,37]. A solution composed of 9 mM Na_2_CO_3_ was used at a flow rate of 1.0 mL min^−1^, and each chromatographic run was completed in 20 min. The system was interfaced with a personal computer for data acquisition/processing and instrumentation control, via a proprietary network chromatography data system (Chromeleon 7.2.8—Thermo Scientific).

The analytical determination of nitrite and nitrate by ion chromatography with conductivity detection is internationally recognized as the most effective approach for this type of analysis. However, to maximize the reliability of the obtained findings, especially the most significant, this technique was coupled to another alternative, based on capillary ion chromatography. All the samples with nitrate concentrations higher than the legal limits, and/or with nitrite concentration higher than 50 mg kg^−1^, were also analyzed by using this alternative method, based on the same procedure of sample preparation, but a chromatographic separation based on gradient elution of diluted KOH [38]. For these confirmation analyses, a Dionex ICS-4000 capillary HPIC system (Thermo Fisher Scientific), composed of a dual-stepper motor pump, an EGC KOH eluent generator cartridge, a 4-port high-pressure valve with 0.4-μL internal sample loop, an ACES 300 anion capillary electrolytic suppressor cartridge set at 13 mA, a column heater set at 30 °C, an EG degas cartridge, a compartment set at 15 °C, and a conductivity detector with cell heater set at 35 °C were used. The chromatographic separations were carried out by using a Dionex IonPac AS11-HC capillary column (250 mm × 0.4 mm i.d.; Particle size, 9 µm, Thermo Fischer Scientific), coupled to the Capillary Guard IonPac AG11-HC (50 mm × 0.4 mm i.d., Thermo Fischer Scientific). The system was interfaced via proprietary network chromatographic software (Chromeleon 7, Thermo Fisher Scientific) to a personal computer for instrumentation control, data acquisition, and processing.

Both analytical methods were fully validated according to the most representative references and guidelines [32,39,40,41,42,43,44], accredited by the Italian Body for Laboratory Accreditation ACCREDIA, submitted to annual proficiency test round, and routinely adopted for the official control of nitrite and nitrate in food. It is worth mentioning that both techniques were employed within the certification procedure of a reference material (nitrate in spinach powder), in 2018–2019 [45]. The most important validation parameters that characterize the two analytical methods used in this monitoring are summarized in Table 1 [30,34].

### 2.4. Statistical Analysis

A value corresponding to LOQ/2 of the analyte was assigned when the concentration of nitrite and/or nitrate was lower than the respective LOQ. These values were used for data elaboration. This is the approach, named “middle-bound” and indicated by the Italian Institute of Health in the document “Rapporti ISTISAN 04/15”, which represents a protective measure for human health and the environment [46].

The statistical analysis was developed to investigate the difference among the levels of these two analytes in different leafy vegetables and, within the same product type, between different packaging and storage conditions (fresh, frozen, and fresh-cut). This comparison was carried out by using the one-way ANOVA and the t-test with a confident interval of 95% (*p* < 0.05), both performed by using Microsoft Excel software.

A contribution to risk assessment was also provided. The acceptable daily intakes of nitrite and nitrate, corresponding to 0.07 and 3.7 mg kg^−1^ body weight (b.w.) per day (expressed as ions), respectively, were taken into consideration for risk-exposure estimation, under a probabilistic approach [47,48].

The same approach used in a recently published paper focused on nitrite and nitrate in meat products was used for risk exposure evaluation [49]. This can be useful for a global-intake evaluation. The low- and high-exposure scenarios were taken into consideration using, for each product type, the lowest and highest concentration levels of nitrite and nitrate registered during monitoring, respectively. Regarding vegetable consumption, the reference data provided to EFSA by the INRAN-SCAI 2005-06 Italian National surveys on food consumption were used, considering the mean data [50,51]. The Margin of Safety (MoS) was evaluated considering that nitrite and nitrate are not genotoxic/carcinogenic, thus, 1 was used as the minimum requirement (bigger is better) and 100 as a protective threshold [52]. The MoS was calculated as the acceptable daily intake (ADI)/estimated exposure dose ratio, under low- and high-exposure scenarios. The ADI values were elaborated for both toddlers and adults, considering 12 and 70 kg as reference b.w., respectively, as established by EFSA [7,48,53].

## 3. Results and Discussion

The results obtained by analyzing 124 samples of leafy vegetables are summarized in Table 2. From an analytical point of view, the first important result to underline is the substantial accordance between the concentrations of nitrite and nitrate detected by analyzing the samples using two different chromatographic separations, as described earlier. This means that the concentrations detected using the two techniques were superimposable, since the quantification obtained from the confirmatory analysis was always in the range covering the HPIC result ± measurement uncertainty (see Table 1). As an example, two chromatograms related to both Swiss chard and wild rocket samples, together with the obtained concentrations, are shown in Figure 2 and Figure 3 relating to HPIC and CIC, respectively. Overall, the nitrite concentrations detected in the following samples were confirmed using both techniques, since they were higher than 50 mg kg^−1^ f.w.: one sample of Swiss chard and three samples of wild rocket (both fresh-cut). Regarding nitrate, three samples of Swiss chard (fresh-cut) were confirmed, since the concentration, higher than 5000 mg kg^−1^, was considered worthy of attention. Moreover, all wild rocket samples with nitrate concentrations higher than the legal limit (6000 mg kg^−1^, since the samples were harvested between 1 April and 30 September) were confirmed, as they were non-compliant. Regarding statistical analysis of both nitrite and nitrate, through the Shapiro–Wilk test, performed by using Microsoft Excel software, data normality was not confirmed for datasets related to different types of vegetables. This is largely justifiable, considering that, as seen above, the accumulation of these compounds in leafy vegetables is influenced by several factors.

### 3.1. Nitrite Determination

The ANOVA one-way and t-test revealed that no statistically significant difference (*p* > 0.05) existed between the nitrite levels detected in the Swiss chard and wild rocket samples, even if collected as fresh, frozen, or fresh-cut. Quantifiable concentrations of nitrite (>LOQ: 4.5 mg kg^−1^) were detected in 37 samples (23 Swiss chard and 14 wild rocket, corresponding to 29% and 32% of analyzed samples, respectively). Most concentrations detected during the survey (33 out of 37) were lower than 50 mg kg^−1^ (highest and mean concentrations corresponded to 43.6 and 14.8 mg kg^−1^, respectively). These low levels have previously been detected in other investigations on leafy vegetables developed worldwide [54,55,56], though it is very difficult to find specific data related to Swiss chard and wild rocket samples [57]. However, the presence of low nitrite concentration should still be considered regarding food safety concerns. Indeed, despite the detection of low “natural” levels of small compounds with preservative effects (e.g., nitrate, sulphite) in food is not uncommon, the legislature has not provided specific legal limits so far [36,58]. This is mainly due to the lack of evidence of toxicological effects of such compounds at low levels. Consequently, this study focused on samples with high concentrations of nitrite (>50 mg kg^−1^). In this survey, these high concentrations were detected in four samples: one Swiss chard (131.6 ± 13.0 mg kg^−1^) and three wild rocket (66.6 ± 6.6, 206.6 ± 20.4, and 219.5 ± 21.7 mg kg^−1^). The most important finding to underline is related to the packaging type of these samples, since all were of the fresh-cut type. Thus, it is possible to hypothesize the role of specific conditions of fresh-cut products in the reduction of nitrate to nitrite, in particular, the temperature of storage (refrigeration), the scarce availability of oxygen, and the cutting operations made before packaging. This last operation could cause the release of extra- and intra-cellular juices in the product, which favors microbial growth, including that of nitrate-reducing bacteria. New research is needed for identifying such mechanisms of nitrite formation in leafy vegetables, considering that similar results were also reported for spinach and lettuce [26]. The identification of nitrite sources in vegetables would be very useful at the industrial level, for standardizing the good manufacturing practices and HACCP aimed at avoiding the reduction of nitrate to nitrite.

Another important aspect to underline is the absence of information about the possible formation of N-nitroso compounds in leafy vegetables with a high content of nitrite. Taking into account that the amines, such as dimethylamine, diethylamine, pyrrolidine and piperidine, which are precursors of the most potent pro-carcinogenic nitrosamines, are present in the free form in vegetables, up to ppm levels, and that also biogenic amines can be found at not negligible levels, this food safety aspect is worthy of more research [59,60]. It is also worthy of mention that some types of leafy vegetables can be processed under acidic conditions (pickled and/or added with acidity regulators), so that a part of the high natural concentration of nitrate might be reduced to nitrite. Thus, further investigation on the N-nitrosamines level in leafy vegetables, especially if processed, is needed.

Another final result of this part of the investigation is the need for legal limits for nitrite in leafy vegetables. Indeed, in two cases, the concentration detected was higher than the legal limit established in Regulation No. 1333/2008/EC for meat products (150 mg kg^−1^) [61].

The results on risk exposure are reported in Table 3. The first evaluation was made regarding the minimum requirement of safety, which is a MoS value of 1. Only the MoS value related to fresh-cut Swiss chard consumed by toddlers under the high-exposure scenario (0.9) was less than 1.0. Regarding the protective MoS value of 100, only 8 values out of 24 (33.3%) had higher results. For toddlers, these higher values were obtained only for fresh and fresh-cut wild rocket samples under the low-exposure scenario (104.5 and 108.5, respectively). Obviously, the general framework is more reassuring for adults, where six MoS values were higher than 100. However, a very low value, worthy of attention, was obtained for fresh-cut Swiss chard under the high-exposure scenario (5.2). This is another important finding in the view of establishing legal limits for nitrite in leafy vegetables, particularly Swiss chard and wild rocket.

### 3.2. Nitrate Determination

The ANOVA one-way and t-test revealed a statistically significant difference (*p* < 0.05) between the nitrate levels detected in Swiss chard and wild rocket samples. In particular, the highest concentrations were quantified in the fresh-cut wild rocket samples (mean concentration: 5043 mg kg^−1^), followed by other categories (mean concentrations in the range 2104.6–3639.3 mg kg^−1^). Significantly lower concentrations (*p* < 0.05) were detected in ribbed Swiss chard samples (mean concentration: 599.7 mg kg^−1^), including two samples with no quantifiable concentration of nitrate (<LOQ: 9.6 mg kg^−1^). This result confirmed that low amounts of nitrate may accumulate in ribbed Swiss chard compared with Swiss chard, as it has been reported that the nitrate accumulation in the midrib is lower than in the leaf blade and petiole [26]. Regarding Swiss chard samples, the mean concentration detected by analyzing 67 samples, collected as fresh, frozen, and fresh-cut, was 2522.6 mg kg^−1^. This mean level is slightly higher than those reported in the literature related to other leafy vegetables (spinach and lettuce), for which legal limits have been defined. For instance, in spinach and lettuce samples, mean concentrations in the range of 613.5–2508.0 and 559.0–2841.0 mg kg^−1^, respectively, were reported [28,62,63,64]. This finding is significant, since considering the nitrate occurrence and the comparable mean consumption of such leafy vegetables (7.1, 17.5 and 0.6 g d^−1^ for Swiss chard, lettuce, and spinach, respectively [50]), the lack of legal limits for nitrate in Swiss chard would seem unmotivated. In comparison with the few data available in the literature for nitrate in Swiss chard, the mean concentration obtained during this monitoring was higher than that reported by Brkić et al. (972.0 mg kg^−1^) [31], but very similar to that reported by Santamaria et al. [30]. As mentioned below, the higher levels quantified in the present monitoring could have been due to the greater concentrations detected in fresh-cut samples, which were not investigated in the other surveys available in the literature. Moreover, it is worth of remembering that nitrate accumulation in leafy vegetables is highly correlated with the specific amount of nitrogen fertilizers applied, which is extremely variable [28,31].

Eleven samples of wild rocket were classified as non-compliant (Table 2). Taking into account the legal limit defined in the European Commission Regulation (EU) No. 1258/2011 for wild rocket harvested from 1 April to 30 September (6000 mg kg^−1^), the higher concentrations quantified in fresh (four samples) and fresh-cut (seven samples) wild rocket were in the range 6478.3–7311.2 and 6042.7–7206.4 mg kg^−1^, respectively. From a legislative point of view, taking into consideration the measurement uncertainty of the method (12.0%), only six samples can be confirmed as non-complaint: three samples of fresh wild rocket (6904.9 ± 828.6, 7302.7 ± 876.3, 7311.2 ± 877.3 mg kg^−1^) and three samples of fresh-cut wild rocket (7206.4 ± 864.8, 6899.0 ± 827.9, 7012.9 ± 841.5 mg kg^−1^).

Regarding risk exposure, as shown in Table 3, no MoS value was less than 1. However, it is worth mentioning that three MoS values were close to 1. In these cases, the nitrate exposure of toddlers related to fresh, frozen, and fresh-cut Swiss chard, under the high-exposure scenario, resulted in MoS values equal to 1.7, 1.6, and 1.1, respectively. Regarding the protective MoS value of 100, only 4 values out of 24 (16.7%) were higher. For toddlers, only the MoS related to ribbed Swiss chard under the low-exposure scenario was higher than 100 (191.8). The general framework is slightly more reassuring for adults, where three MoS values were higher than 100. However, several MoS values were very low, such as in the case of fresh-cut Swiss chard consumption under the high-exposure scenario (6.3). These results are very important regarding updating/establishing legal limits for nitrate in Swiss chard and wild rocket. Indeed, the concern raised from this risk exposure suggests both the introduction of legal limits for nitrate in Swiss chard and nitrite in both Swiss chard and wild rocket and the intensification of official controls, especially regarding the levels of nitrate in wild rocket. The greater mean concentrations of both nitrite and nitrate revealed in the Swiss chard and wild rocket samples commercialized as fresh-cut products are also a significant result. This finding suggests the development of further research aimed at identifying and evaluating the causes of such increases in these products. These studies could be very useful at the processing-industry level, to avoid the presence of high concentrations of nitrite/nitrate in fresh-cut leafy vegetables, improving the food safety level of such products, significantly.

## 4. Conclusions

In this study, two particular types of leafy vegetable, Swiss chard and wild rocket, were investigated with the aim of evaluating their levels of nitrite and nitrate. The analyses were carried out by using validated and accredited ion chromatography with conductivity-detection methods.

Regarding nitrite, high concentrations, in the range 66.6–219.5 mg kg^−1^, were detected in one Swiss chard and three wild rocket samples. The MoS, evaluated for toddlers and adults under low- and high-exposure scenarios, was <1 in one Swiss chard sample, and 67% of values were <100. Regarding nitrate, in Swiss chard samples the mean concentration (2522.6 mg kg^−1^) was slightly higher than those reported in the literature for spinach and lettuce. Regarding risk exposure, no MoS was <1, but 83% of values were <100. Nitrate concentrations higher than the legal limit were quantified in 11 wild rocket samples. The survey and related risk-exposure study confirmed the need for official control, suggesting the introduction of legal limits for nitrate in Swiss chard and nitrite in both Swiss chard and wild rocket.

## Figures and Tables

**Figure 1 foods-11-02571-f001:**
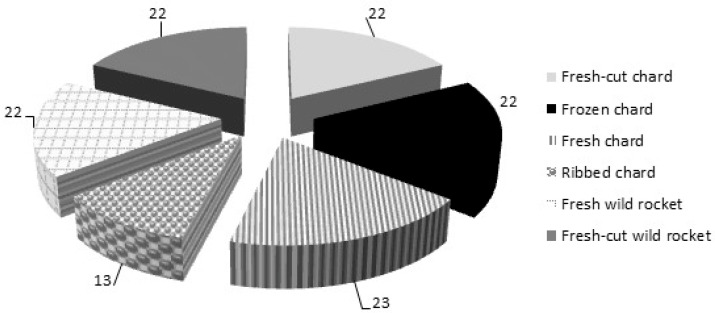
Number and type of samples analysed in this study.

**Figure 2 foods-11-02571-f002:**
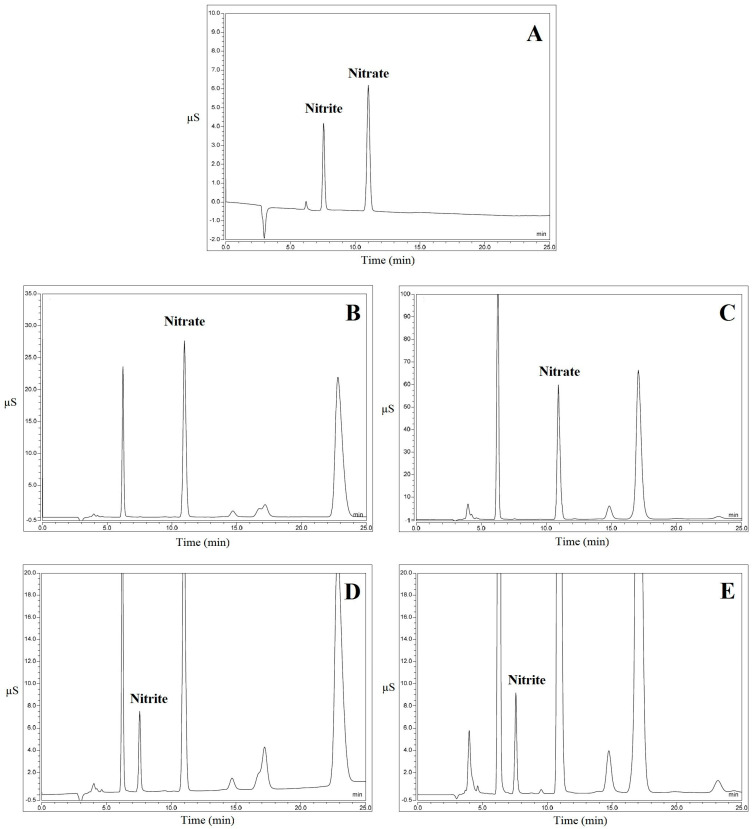
Chromatogram examples of HPIC analyses. (**A**): Nitrite and nitrate standard solution; (**B**): Fresh-cut Swiss chard sample with nitrate concentration of 2838.8 mg kg^−1^; (**C**): Fresh-cut wild rocket sample with nitrate concentration of 7012.9 mg kg^−1^; (**D**): Fresh-cut Swiss chard sample with nitrite concentration of 131.6 mg kg^−1^; (**E**): Fresh-cut wild rocket sample with nitrite concentration of 206.6 mg kg^−1^.

**Figure 3 foods-11-02571-f003:**
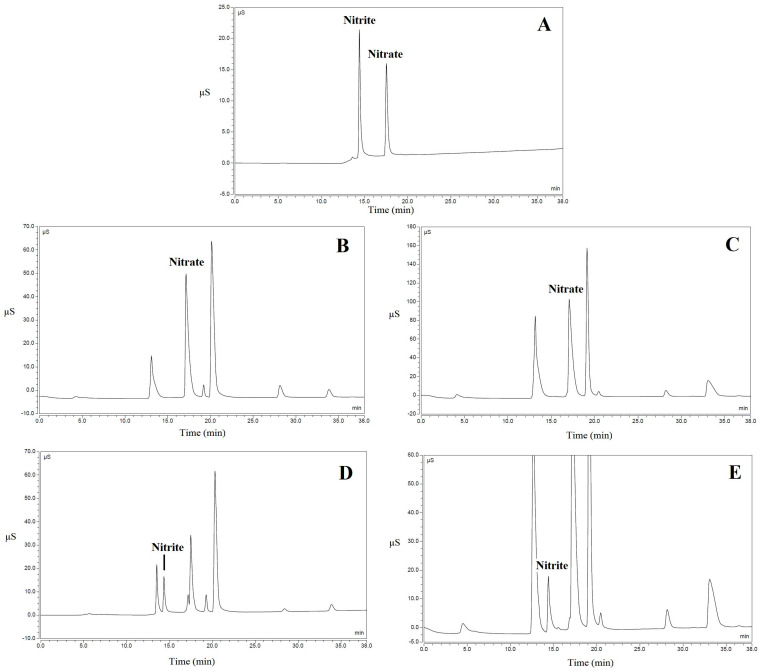
Chromatogram examples of CIC analyses. (**A**): Nitrite and nitrate standard solution; (**B**): Fresh-cut Swiss chard sample with nitrate concentration of 3096.4 mg kg^−1^; (**C**): Fresh-cut wild rocket sample with nitrate concentration of 6471.5 mg kg^−1^; (**D**): Fresh-cut Swiss chard sample with nitrite concentration of 145.5 mg kg^−1^; (**E**): Fresh-cut wild rocket sample with nitrite concentration of 191.4 mg kg^−1^.

**Table 1 foods-11-02571-t001:** Main validation parameters of high-pressure ion chromatography (HPIC) and capillary ion chromatography (CIC).

Analytical Technique	Analyte	Linearity (R^2^)	Limit of Detection *	Limit of Quantification *	Mean Precision (%) (n = 18)	Mean Recovery (%) (n = 18)	Selectivity and Application Field	Measurement Uncertainty (%)
HPIC	Nitrite	0.999	1.5	4.5	4.1	98.7	Meat, feed, cheese products, seafood, leafy vegetables	9.9
	Nitrate	0.999	3.2	9.6	4.2	98.3		12.0
CIC	Nitrite	0.999	0.7	2.1	3.4	98.8	Meat, fish, vegetables, cereals and pulses, feed, cheese products, seeds and nuts	2.6
	Nitrate	0.999	1.2	3.7	2.7	100.8		2.4

* Expressed as mg kg^−1^ of sample.

**Table 2 foods-11-02571-t002:** Nitrite and nitrate levels (mg kg^−1^) and distribution (%) in different leafy vegetables (n = 124).

Sample Type	Number of Samples Analyzed	Number of Samples with Concentration < LOQ *	Mean Concentration (mg kg^−1^) ***	Median (mg kg^−1^)	Concentration Range (mg kg^−1^) **	Number of Non-Compliant Samples
			NITRITE			
Fresh chard	23	17 (74%)	5.0	2.25	7.0–18.3	-
Frozen chard	22	16 (73%)	5.0	2.25	6.0–19.5	-
Fresh-cut chard	22	14 (64%)	11.5	2.25	6.5–131.6	-
Ribbed chard	13	10 (77%)	3.3	2.25	4.5–9.5	-
Fresh wild rocket	22	17 (77%)	5.8	2.25	13.4–20.1	-
Fresh-cut wild rocket	22	13 (59%)	30.2	2.25	12.9–219.5	-
Total	124					
			NITRATE			
Fresh chard	23	0 (0%)	2104.6 ^b^	2104.6	367.9–3764.9	-
Frozen chard	22	0 (0%)	2418.9 ^b^	2370.2	1057.0–3968.4	-
Fresh-cut chard	22	0 (0%)	3063.2 ^a,b^	3123.4	442.2–5834.9	-
Ribbed chard	13	2 (15%)	599.7 ^c^	503.8	32.6–2150.3	-
Fresh wild rocket	22	0 (0%)	3639.3 ^a,b^	3638.3	1121.6–7311.2	4 (18%)
Fresh-cut wild rocket	22	0 (0%)	5043.0 ^a^	4702.6	3517.7–7206.4	7 (32%)
Total	124	2 (2%)	2966.3	2922.1	32.6–7311.2	-

* LOQ = 4.5 mg kg^−1^ (nitrite), 9.6 mg kg^−1^ (nitrate). ** Evaluated only for samples with concentration > LOQ. *** Means with different superscript (a, b, c) significantly differ (*p* < 0.05).

**Table 3 foods-11-02571-t003:** Margin of Safety (MoS) for chard and wild rocket consumption under low- and high-exposure scenarios.

Toddlers (b.w. 12 kg)							
		Fresh Chard	Frozen Chard	Fresh-Cut Chard	Ribbed Chard	Fresh Wild Rocket	Fresh-Cut Wild Rocket
Nitrite	**Low exposure**	16.9	19.7	18.2	26.3	104.5	108.5
	**High exposure**	6.5	6.1	**0.9**	12.5	69.7	6.4
Nitrate	**Low exposure**	17.0	5.9	14.1	191.8	66.0	21.0
	**High exposure**	1.7	1.6	1.1	2.9	10.1	10.3
**Adults (b.w. 70 kg)**							
		Fresh Chard	Frozen Chard	Fresh-Cut Chard	Ribbed Chard	Fresh Wild Rocket	Fresh-Cut Wild Rocket
Nitrite	**Low exposure**	98.6	115.0	106.2	153.4	609.5	633.1
	**High exposure**	37.7	35.6	5.2	72.6	406.3	37.2
Nitrate	**Low exposure**	99.2	34.5	82.5	1119.0	384.9	122.7
	**High exposure**	9.7	9.2	6.3	17.0	59.0	59.9

In bold, the MoS value is lower than 1.0. Mean consumptions available from: INRAN-SCAI 2005-06. https://www.crea.gov.it/documents/59764/0/appendice_3b5_verdura.pdf/f32e7f5d-d7ad-69c0-4d76-ffd894e7ca2c?t=1550827463788, accessed on 21 August 2022.

## Data Availability

The data presented in this study are available on request from the corresponding author.

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
