# Peer review of "Determination of Nitrate and Nitrite in Swiss Chard (Beta vulgaris L. subsp. vulgaris) and Wild Rocket (Diplotaxis tenuifolia (L.) DC.) and Food Safety Evaluations"

_foods, 2022, doi:10.3390/foods11172571_

Round 1
Reviewer 1 Report
In my opinion, the manuscript covers very important topic and shows interesting and important results concerning nitrates and nitrites in two species of widely consumed leafy vegetables. It is also generally well-written, although in my opinion some corrections are necessary. Below I have provided some specific suggestions for improvements (mostly minor) and a few questions to the Authors (to be addressed in the manuscript).
General technical aspects:
- first letter of each word in the manuscript’s title & the sections- and subsections’ titles should be capital.
- spaces between paragraphs should be removed (adequate style from the Foods Journal template should be applied to the text of the manuscript.
Introduction:
Line 27-29: Appropriate references (citations) should be given here.
Lines 30-34: This paragraph suggests that the nitrates and nitrites measured in the tested plants could come from the listed sources (chemical industry, explosives etc. releasing nitrates to the environment) and not from plant metabolism. If Authors would like to make such a statement, relevant proofs/references should be provided.
Lines 39-43: I would suggest to include also some discussion around positive effects of nitrates reported in many studies.
Lines 44-52: The aspect of formation of nitrosamines in human stomach should be also mentioned/described (with relevant references).
Methods:
Lines 92-95: What was a size of a single sample? Were they all conventional, or also organically produced? I would suggest to add this information.
Line 96: Where are the “local stores” located? (a brief location of the study/sampling area should be mentioned). What was the origin of the vegetables? (cultivation area/at least brief)?
Line 104: I suggest consistency in using spaces between temperature (number) and unit (Celsius degrees) throughout the manuscript.
Line 112: (1000 mg L-1) – “-1” should be in superscript. The same comment refers to Line 120.
Lines 132-133: “The isocratic chromatographic separation is well-known and standardized for meat products analysis” – in such case is this standardization relevant for the leafy vegetables tested?
Line 177: mg/kg should be changed into mg kg-1 (for consistency of using always the same form throughout the manuscript).
Lines 165-192 (statistical analysis): What was the software used for the statistical analyses? Were data normally distributed? How was the normality checked? These information should be provided.
Results and Discussion:
All 3 tables should be placed directly in the manuscript (and not as ‘supplementary files’).
References:
List of references needs more attention to strictly follow the Journal’s requirements (e.g. some journal names are not abbreviated).
Author Response
The authors would like to thank the reviewer for his effort in improving the scientific impact of the Paper. The manuscript has been revised, according to reviewer’s suggestions, editing corrections and rewording the text where necessary. Please note that the references to lines are referred to the word version of the revised article.
Reply to Reviewer 1
In my opinion, the manuscript covers very important topic and shows interesting and important results concerning nitrates and nitrites in two species of widely consumed leafy vegetables. It is also generally well-written, although in my opinion some corrections are necessary. Below I have provided some specific suggestions for improvements (mostly minor) and a few questions to the Authors (to be addressed in the manuscript).
General technical aspects:
- first letter of each word in the manuscript’s title & the sections- and subsections’ titles should be capital.
Response: Corrected.
- spaces between paragraphs should be removed (adequate style from the Foods Journal template should be applied to the text of the manuscript.
Response: Corrected.
Introduction:
Line 27-29: Appropriate references (citations) should be given here.
Response: The first two references have been added for this section.
Lines 30-34: This paragraph suggests that the nitrates and nitrites measured in the tested plants could come from the listed sources (chemical industry, explosives etc. releasing nitrates to the environment) and not from plant metabolism. If Authors would like to make such a statement, relevant proofs/references should be provided.
Response: This section is focused on nitrate only. At lines 66-70, some references have been added about the possible presence of high levels of nitrite in vegetables and about its possible origin from nitrate reduction. Unfortunately, to date, no study is available regarding this phenomenon in vegetables intended for human consumption.
Lines 39-43: I would suggest to include also some discussion around positive effects of nitrates reported in many studies.
Response: Thanks for your suggestion. Two references and new comments have been added at lines 57-59.
Lines 44-52: The aspect of formation of nitrosamines in human stomach should be also mentioned/described (with relevant references).
Response: Following the reviewer’s suggestion, a new comment has been added at lines 44-49, together with the original reference.
Methods:
Lines 92-95: What was a size of a single sample? Were they all conventional, or also organically produced? I would suggest to add this information.
Response: Thanks for your comment. This information has been added at lines 108 and 114-115.
Line 96: Where are the “local stores” located? (a brief location of the study/sampling area should be mentioned). What was the origin of the vegetables? (cultivation area/at least brief)?
Response: According to the referee’s remark, an overview of sample origin has been added at lines 103-108.
Line 104: I suggest consistency in using spaces between temperature (number) and unit (Celsius degrees) throughout the manuscript.
Response: Corrected.
Line 112: (1000 mg L-1) – “-1” should be in superscript. The same comment refers to Line 120.
Response: Corrected (now lines 125 and 133). The same correction was also made at line 224.
Lines 132-133: “The isocratic chromatographic separation is well-known and standardized for meat products analysis” – in such case is this standardization relevant for the leafy vegetables tested?
Response: Thanks for this remark. The sentence at lines 146-147 has been completed, specifying method applicability for vegetables analysis, also adding another relevant reference (n.34).
Line 177: mg/kg should be changed into mg kg-1 (for consistency of using always the same form throughout the manuscript).
Response: Corrected (now line 195).
Lines 165-192 (statistical analysis): What was the software used for the statistical analyses? Were data normally distributed? How was the normality checked? These information should be provided.
Response: Thanks for your suggestion. This info about statistical analysis has been added both at lines 193 and 227-231.
Results and Discussion:
All 3 tables should be placed directly in the manuscript (and not as ‘supplementary files’).
Response: As suggested, the 3 tables have been added in the text, where needed.
References:
List of references needs more attention to strictly follow the Journal’s requirements (e.g. some journal names are not abbreviated).
Response: Following the reviewer’s remark, the abbreviations in references section have been carefully checked and corrected at line 488 and for some missing punctuations. Unfortunately, no abbreviation is available for some journals, such as Pharmacology and Pharmacy, International Journal of Hypertension etc., from the Web of Science database.

Reviewer 2 Report
The article presents the investigation of nitrite and nitrate levels in Swiss chard and wild rocket by using validated analytical methods. The text is clear and precise, and the data obtained are valuable and relevant to the scientific community.
Below are some suggestions to the authors:
Abstract:
- Consider removing the sentence. “Some authors also reported possible high nitrite levels”.
- Materials and methods:
- Specify the city or cities where samples were collected
Results and discussion:
- Consider rewriting the phrase “From an “analytical” point of view…” line 196,and “… from the “confirmatory” analysis…”, line 200. Or, remove the quotation marks. The same for “confirmed” in line 204-205 and “non-compliant” in line 210.
Author Response
The authors would like to thank the reviewer for his effort in improving the scientific impact of the Paper. The manuscript has been revised, according to reviewer’s suggestions, editing corrections and rewording the text where necessary. Please note that the references to lines are referred to the word version of the revised article.
Reply to Reviewer 2
The article presents the investigation of nitrite and nitrate levels in Swiss chard and wild rocket by using validated analytical methods. The text is clear and precise, and the data obtained are valuable and relevant to the scientific community.
Below are some suggestions to the authors:
Abstract:
- Consider removing the sentence. “Some authors also reported possible high nitrite levels”.
Response: As suggested by the reviewer, the sentence has been removed.
- Materials and methods:
- Specify the city or cities where samples were collected
Response: Thanks for your suggestion. This info has been added at lines 103-108.
- Materials and methods:
Results and discussion:
- Consider rewriting the phrase “From an “analytical” point of view…” line 196, and “… from the “confirmatory” analysis…”, line 200. Or, remove the quotation marks. The same for “confirmed” in line 204-205 and “non-compliant” in line 210.
Response: As suggested by the reviewer, quotation marks have been removed (lines 134, 159, 213, 217, 221, 227, 334, 340 and table 2).
